# Parents’ Perceptions of Educational Apps Use for Kindergarten Children: Development and Validation of a New Instrument (PEAU-p) and Exploration of Parents’ Profiles

**DOI:** 10.3390/bs11060082

**Published:** 2021-05-27

**Authors:** Julie Vaiopoulou, Stamatios Papadakis, Eirini Sifaki, Dimitrios Stamovlasis, Michail Kalogiannakis

**Affiliations:** 1Department of Education, University of Nicosia, 2417 Nicosia, Cyprus; vaiopoulou.cp@unic.ac.cy; 2Department of Preschool Education, Faculty of Education, University of Crete, 74100 Crete, Greece; mkalogian@uoc.gr; 3Department of Cultural Organisations Management, Hellenic Open University, 26222 Patras, Greece; sifaki.eirini@ac.eap.gr; 4Department of Philosophy and Education, Aristotle University of Thessaloniki, 54124 Thessaloniki, Greece; stadi@auth.gr

**Keywords:** educational apps, parent’s perception, PEAU-p, LCA, parents’ profiles

## Abstract

Contemporary mobile technologies offer tablets and smartphones that elicit young children’s active participation in various educational apps, dramatically transforming playing, learning, and communication. Even the most knowledgeable users face difficulties in deciding about the value and appropriateness of the so-called educational apps because of many factors that should be considered. Their importance for children’s attitudes is affected by the perceived positive and negative aspects, which vary across a multiplicity of criteria. Filling the gap in the relevant literature, a new instrument, named PEAU-p (Perceptions about Educational Apps Use–parents), was developed and validated in the present study designed to measure parents’ perception of educational apps for kindergarten pupils. Data (*N* = 435) were collected via online procedures, and the psychometric properties of PEAU-p were studied via exploratory and confirmatory methods. Principal Components Analysis extracted six factors, namely Usability, Enjoyment, Involvement, Learning, Worries, and Values, which explained 72.42% of the total variance. Subsequently, by implementing Latent Class Analysis based on the above factors, four Clusters (i.e., parents’ Profiles) were extracted corresponding to their perceptions and attitudes towards the educational apps used for kindergarten pupils. Those were named as ‘mild attitude’, ‘negative attitude’, ‘positive attitude’, and ‘indifferent attitude’. This categorization, besides the statistical support, is fully interpretable, and the profiles were associated with certain covariates, such as age, the number of children, knowledge on new technologies, or distal outcomes, e.g., the frequency of using apps, the general position towards apps or their intention to recommend apps use. The findings are discussed within the current research field, investigating the influential role parents play in young children’s media selection and use.

## 1. Introduction

Since Apple launched iPhone in 2007 and the iPad in 2010, people worldwide have been attracted to and fascinated by touchscreen devices [1,2]. Touchscreen devices provide an intuitive and straightforward method of management. Compared to the customary systems, the interactive screens did not require special motor skills and became easy to use even for young children [3]. Studies with young children have shown that basic operations, such as opening apps, and tracing shapes, or swiping the screen, are efficiently utilized [4]. These devices also offer the advantage of speed, ease of learning, and flexibility [5].

The learning advantages of smart mobile devices have been acknowledged by many scholars and researchers [6], and comparisons with the usage of traditional devices showed that the new interactive units provide the opportunity for more efficient learning [7,8,9]. Preschoolers might benefit specifically in areas such as the development of basic skills (reading, writing, and mathematics), improvement of basic cognitive and emotional skills, support of collaboration, etc. [10,11,12]. Studies employing experimental designs with control groups provided empirical evidence for the positive impact in math achievement when using apps at home [13,14], when compared to standard classroom practice [15], or even when compared to placebo control, i.e., a non-educational app [16]. Recent reviews on the impact of educational apps (e.g., [17]) have shown benefits not only on mathematics but also on literacy development, science, problem-solving, and self-efficacy.

Using apps in a fully interactive learning environment is stimulating for children who might be tired of the typical learning model [18]. When children use them, they may experience fun and joy while simultaneously facing challenging activities, exploring unknown territories, and being creative by producing new texts expressing their ideas and thoughts and connecting them with real-life [19]. Acknowledging this, the American Academy of Pediatrics, when referring to the mobile app technology, endorsed its usage and highlighted that it is essential and necessary for children and parents to be engaged together [20]. It is pertinent to mention at this point that the positive findings mentioned above are not fully supported in the case of children with autism spectrum disorder (ASD) (e.g., [21]).

In many parts of the world, children’s use of media has become part of a “family-based media ecology” ([22], p. 7) since both children and parents participate in everyday home activities implementing touch screen devices [3,17,23]. Over 50% of the educational apps available in the market are offered to preschool pupils, while parents recognize their educational value, approve app usage, and seek to purposefully and meaningfully incorporate touchscreens into simple literacy instruction [24,25,26,27]. American research has shown that the number of children below eight years old using mobile media at home became three times larger during 2013–2017, and this increase is anticipated to continue. Comparable estimates were reported in developed countries globally [4,27]. A survey in the USA indicated that 90% of toddlers below two years old have experienced playing with a touchscreen, and 73% of children between five and eight years old have been using tablets regularly. In a study of low-income families in the United States, it was found that 75% of the children possess and use mobile devices by the age of four, while it was reported by Common Sense Media (2013) that nearly 60% of the parents in the USA installed educational apps for their children [28,29,30].

### 1.1. The Parental Role in Preschooler Educational App Usage 

Parents and teachers are looking for such resources that offer play, entertainment, and learning; on the other hand, designers try to satisfy those needs and supply the market with interactive content apps for children [31]. Parents of preschoolers acting as ’media gatekeepers’ play a crucial role in children–app interactions because they decide about the digital technology selection, the type, and the frequency of its use. Their permission or conditional consent significantly affects the child’s attitude and engagement within the ensuing interaction process. It is acknowledged, given the integrated home technologies, that when parents make appropriate choices and encourage children to be engaged with high-quality apps, they could scaffold children’s cognitive development, are imaginary, and learn via a self-driven inquiry [17,32,33,34,35]. Research on parents and app usage at home indicated that it could be beneficial for all family members, offering rich interactive material, endowing parents’ awareness, and promoting their skills to effectively expedite their children’s experiences. Thus, parents are responsible for the choices, and they should be cautious, avoiding those digital products that usually do not promote creativity [36,37].

Even though parents’ involvement in selecting and implementing preschool children apps is crucial, they frequently use them without accurately evaluating their educational quality or their usefulness, an attribute highly dependent on their developmental process and design. Indeed, despite their popularity, some of the products are not suitable for young ages, and, in addition, they do not offer the anticipated prospects for creativity and enjoyment either [34,38,39].

Beyond the advantages of apps mentioned above, even the top products that supposedly cover reading and mathematics ability, such as those available in the Apple and Android App Store, enclose a mix of low- and high-quality aspects. Thus, as challenging it may seem, it is the parents’ responsibility to identify the ones that are suitable for use at home and to distinguish them from those that are ill-designed and do not take advantage of all the hardware and software capabilities provided by a smart screen device [40,41,42].

Even though most parents and caregivers use touchscreen devices for themselves, they may face difficulties implementing them with their young children, as often they ignore the appropriate manners to encourage children and the way to attain early learning development at home. This was shown irrespective of the parental views about the digital play of their children. Parents’ knowledge was proved inadequate to support digital game use, and their effects on desired outcomes are rather vague [4,43,44,45,46,47,48,49]. However, it was found that parents focus on particular aspects of apps. For example, it is less likely that the parents think about specific features of educational apps, such as visual and sound effects, than to consider the actual content [50]. The above literature shows that parental perceptions about educational apps are decisive for their use by children because parents determine what types of mobile devices will be in use.

### 1.2. Studies Examining Parents’ Perceptions about Educational Apps

In addition, in the shared experience, parents act as advisors and teachers, affecting their attitudes and facilitating learning outcomes. If parents’ perceptions favor fostering apps, positive habits are established, while critical thinking promotes which types of app content is suitable and worth using [51]. The perceptions and behaviors in question, however, are not uniform. Some perceptions are inclined towards selecting apps with educational and learning values, while some others emphasize the value adhered to fun and entertainment [34,50] or seeing these “screen-viewing activities as a babysitter/coping mechanism and a device to wake up or wind down young children” ([52], p. 124). Parents have predominately exhibited five needs as far the apps use is concerned, i.e., to attain acquaintance, pass the time, get entertainment, acquire coeducation, and face personalized challenges [34]. These different needs vary among individuals and might determine or are part of distinct parenting styles, which could be influenced by demographics, the family’s climate, parents’ related knowledge and skills, parents’ general approach to media, and the specific context question [53].

Moreover, other variables have been sporadically reported related to the issue under investigation, such as the marital status and the parents’ education. The higher the education level of parents, the lesser the involvement of children at young ages with screens, while the opposite is observed at lower educational levels, where children are strongly encouraged to be engaged with screen media [1,54]. However, the effects of individual characteristics associated with parents’ perceptions of educational app implementation have not been systematically explored. The initial literature review exhibits a volume of accumulated knowledge, which dictates further research focused on parents’ beliefs and their constructive attitudes on app use, as it could predict their behavior in interaction settings with children [44,55,56,57,58,59]. The reported empirical evidence suggests that the implementation of digital media by both parent and child at home is directly influenced by parental views and perception about digital technology [60]. That is the perceived usefulness and efficacy of apps as entertainment or educational tools to influence the crucial decisions for the availability and accessibility of these technologies at home [55,61]. Among the influential factors, several variables related to emotions, worries, and uncertainty present in parents’ decision-making. This leads to an interesting observation that the parental views in question have expressed as polarized positions, often as strictly positive or negative attitudes towards the implications of the digital technology use for their children’s lives [61,62,63,64]. Frequently, these seemingly opposite standpoints coexist, positing interesting questions for further research. Parents acknowledge the potential benefits of mobile devices [10,64,65,66], but simultaneously, they express some concerns about the possible drawbacks of mobile device use [62,67]. The two opposing attitudes are apparent in everyday discourses. Many parents show enthusiasm for including smart screen technology in the preschool education curricula and acknowledge them as an excellent prospect for preparing young children for their following formal education. On the other hand, other parents emphasize some negative aspects of smart screen technology, and they provide limited opportunities to use it or even prohibit it together [56,62].

Analogously to the past period, research focused on other media, e.g., television, nowadays, educational apps’ experience attracts attention. The research interest has shifted to their implementation and the crucial role of parental involvement [34,68]. The present study investigates this contemporary issue and contributes by presenting an instrument to measure parents’ perceptions of apps and explore their psychometric properties. Moreover, a person-centered approach was implemented, and based on selected individual differences, parents’ profiles were revealed describing their attitudes towards apps’ use. Parents’ perceptions of app users have been accessed previously via qualitative interviews or questionnaires (e.g., [32,50,69]) focusing on various aspects. This work proposes a new scale including dimensions that have not been examined so far and provides its psychometric analysis.

## 2. Materials and Methods

### 2.1. Research Questions

Probing to understand parents’ perceptions and attitudes towards using educational apps by their children primarily requires a valid instrument and a method for measuring them. Parents’ perceptions of apps that create attitudes towards them are probably varying across a few factors and criteria, or the dimensions of the latent variables under study, which finally determines their overall position towards educational apps. In this work, filling a gap in the literature, the new instrument, named PEAU-p (Perceptions about Educational Apps Use—parents), was developed to measure parents’ perceptions of the apps use. Furthermore, it was implemented to explore their constructive attitudes towards them. Based on this initiative and within the premise growing via the abovementioned rich literature, several research questions were formulated: What are the psychometric characteristics of the PEAU-p instrument, including the validity and reliability issues?Based on measured parents’ perceptions on app use: Are there distinct groups/clusters of parents sharing typical profiles reflecting formative attitudes?Are the ensued cluster memberships/profiles associated with other individual differences acting as covariates or distal outcomes?

### 2.2. Participants and Procedures

Participants were 435 parents who completed the PEAU-p questionnaire online. The majority (95.9%) were mothers with ages varying from 21 to 50 years old (median = 35, mean = 36.88, SD = 5.21). The number of children varied from 1 to 4 (mean = 1.68, SD = 1.51). The study was a cross-sectional survey using an electronic sampling process via social media groups. The implemented self-completion questionnaire (PEAU-p) for parents was uploaded on a web-based form via Google Forms, and parents completed it anonymously. The escorting cover letter provided all relevant information, such as elucidating the purpose of the study, the confidentiality of the process, and the voluntary character of the participation, without financial incentives. The protocol was following procedures approved by the Research Ethics Committee from the University of Crete, Greece.

### 2.3. Measures

The PEAU-p is a new instrument designed for measuring parents’ perceptions of app use. The scale was proposed after extensive preliminary work, including studying the relevant literature, theoretical considerations, and the use of valid questions chosen from IT literature, but adopted and refined for the current. Moreover, the new items generated focused on providing the empirical indices for the dimensions conceived to describe the latent variable under investigation. The initially proposed form of PEAU-p was developed via intensive expert-team work and elaboration process that assured, at least at a theoretical level, validity issues, further explored via statistical methods in the next phase.

In PEAU-p, the latent variable under investigation encompasses six dimensions, named: Usability, Enjoyment, Involvement, Learning, Worries, and Values. For the six theoretically conceived dimensions, 27 items were used that remained in the final form of PEAU-p after the exploratory and validation procedures implemented on an initial battery of 39 items.

Besides the PEAU-p items, the questionnaire included two other types of questions: (1) variables that provide demographic information (e.g., gender, age, educational level, number of children, age of children) and (2) several items, operationalizing as relevant attitudinal or cognitive variables, such as, “What is the level of your knowledge on New Technologies?”, “Do you think that kids spend too much time on screens?”, “How often do you use apps with your children?”, “How old was your kid when you start using educational apps?”, “Does the use of apps cause conflicts between you and your children?”, “What is your overall position towards educational apps?” and “How possible is to recommend educational apps to other parents?”. The above items (measured on a 7-point Likert scale) were associated as independent or dependent variables with the parents’ attitudinal profiles ensued from subsequent analyses.

## 3. Results

### 3.1. Exploratory (EFA) and Confirmatory Factor Analysis (CFA)

In the EFA procedure, Principal Components Analysis (PCA) with Varimax rotation was applied. Initially, a rescaled transformation (CATPCA procedure; [70]) was applied to the initial ordinal data to make the resulting scale suitable for PCA. Bartlett’s test of sphericity (χ^2^ = 8045.29, *p* < 0.0001) and the Kaiser–Meyer–Olkin index (0.889) indicated adequate variance for factor analysis. The number of factors was decided based on the screen plot and the Kaiser Criterion, eigenvalue greater than 1. Six factors were extracted. In addition, an auxiliary analysis of Principal Axis Factoring (PAF) and parallel analysis using Monte Carlo simulation resulted in the same dimensionality. A PCA analysis was conducted using an initial pool of 39 items. The refinement with Varimax rotation and keeping only items with loadings more significant than 0.40 resulted in a final interpretable structure using 27 items. The six factors explained 72.42% of the total variance. Table 1 shows the corresponding eigenvalues for Usability (US), Enjoyment (EN), Involvement (IN), Learning (LE), Worries (W), and Values (VA), which are 2.46, 3.45, 1.98, 4.76, 4.12, and 2.80, respectively, while the corresponding portions of variance explained were 9.10%, 12.78%, 7.33%, 17.61% 15.25% and 10.36%, respectively (see Table 1).

Subsequently, confirmatory factor analysis (CFA) for the proposed model was applied to the second part of the data using LISREL8 procedures. The proposed 6-factor structure fitted in a CFA model adequately (χ^2^/df = 1.23, *p* = 0.06, CFI = 0.995, RMSEA = 0.027, 90% CI of RMSEA = [0.000; 0.040], SRMR = 0.07, GFI = 0.90; NFI = 0.957 and NNFI = 0.993). The factorial validity of the proposed structure is sufficiently supported; however, it might be worthwhile mentioning that applying an ESEM (exploratory structural equation modeling) approach that allows cross-loadings in two items, an even better fit is, expectantly, achieved (e.g., [71]).

Table 2 shows the correlation matrix of the six factors, the means, the standard deviations, and the corresponding reliability coefficients. Cronbach’s alphas ranged from 78 to 89, denoting satisfactory internal consistency. These six factors were subsequently used, as input variables, in the LCA procedure.

### 3.2. Latent Class Analysis—Parents’ Profiles

In the next phase, Latent Class Analysis was carried out to reveal distinct groups/clusters or latent classes of participants sharing common profiles. In LCA, the cases/respondents in a latent class are considered homogeneous regarding the model’s parameters that describe the pattern of their responses [72]. The procedure assigns the participants to class/cluster membership based on conditional probabilities characterizing each latent class. The cluster/latent class predictions are achieved via the posterior probability of belonging to a class C, given an observed response pattern y, p(c/y), by applying Bayes’ theorem, which also employs the p(y/c), the conditional probability of y given c, and p(c) and p(y), i.e., the probabilities of c and observed pattern y, respectively [73]. Several indexes assess the latent-class model-fit, i.e., the number of parameters, entropy-R2, likelihood ratio statistic (L2), Bayesian Information Criterion (BIC), Akaike’s Information Criterion (AIC), degrees of freedom, and bootstrapped *p*-value.

LCA is a powerful method having numerous applications in a wide range of disciplines and fields. In educational research, LCA has been proved an effective tool in deriving participants’ profiles and answering challenging research questions. Mentioning some indicative investigations would include exploring students’ predictions and explanations of physical phenomena (e.g., [74]), testing contradictory theoretical perspectives, such as the coherence vs. fragmented knowledge hypotheses (e.g., [75,76,77]). In addition, it should be emphasized that LCA is a psychometric approach and has a series of advantages over the traditional cluster analysis [73].

In the present analysis, the step-wise method was used, which has three steps: (a) the primary latent categorical variable is identified based on a set of indicators, (b) the participants are assigned to latent classes/profiles, and (c) the ensued class membership is associated with covariates or distal outcomes [78,79]. Figure 1 shows the latent variable model with the measurement and the structural parts. In the first step of the LCA, the latent classes/profiles were identified using the six factors of PEAU-p (Usability, Enjoyment, Involvement, Learning, Worries, and Values) as input. In the second step, the parents were allocated to latent classes/profiles utilizing the modal assignment approach, along with the maximum likelihood (ML) bias correction [78]. A detailed illustrative description of the three-step LCA applied to educational research can be found in Stamovlasis, Papageorgiou, Tsitsipis, Tsikalas, and Vaiopoulou [80].

The results of the LCA first step are shown in Table 3. The four-cluster solution had the minimum value of the Bayesian Information Criterion (BIC), and it was chosen as the best parsimonious model (entropy-R^2^ = 0.80, df = 383, classification-error = 0.1320, BIC = 5252.87, Npar = 51).

Figure 2 presents the ensued profiles in terms of conditional probabilities depicted on the vertical axis, while the horizontal axis shows the six dimensions of the PEAU-p scale. Cluster/Profile 1 (31.31%) includes parents with perceived low Usability, Enjoyment, Values and Worries, and medium perceived Involvement and Learning. This was named the Mild Attitude profile. Cluster/Profile 2 (29.07%) includes parents with perceived low Usability, Enjoyment, Values, Learning, but with perceived high Involvement and Worries. This was named the Negative Attitude profile. Cluster/Profile 3 (20.5%) includes parents with medium Worries and perceived high the other five dimensions (Usability, Enjoyment, Values, Involvement, and Learning). This was named the Positive Attitude profile. Cluster/Profile 4 (19.13%) includes parents with low all six perceived dimensions. This was named the Indifferent Attitude profile. Table 4 presents a verbal description of the four profiles.

### 3.3. Association of the Profiles with Other Variables

#### 3.3.1. Effect of Independent Variable on Parents’ Profiles

In the next step, the cluster/profile membership was associated with covariates; the latter considered independent factors affecting the former. The following variables were examined: Parents’ age, number of children, the age of the 1st child, the age of the 2nd child, parents’ knowledge of new technologies, parents’ level of education, and parents’ perception of time spent by children on screens (Table 5). Parents’ age is associated negatively with Profile 2 (Negative Attitude). Parents who do not appreciate educational apps and perceive high involvement and worry about their children’s use are probably the younger ones (b = −0.075, *p* < 0.001). This could be explained by the fact that the younger parents have the younger children; thus, they are directly concerned with exhibiting intense worries. Note that the average age was 36.88 (SD = 5.21). This is in line with the finding that Profile 4, i.e., the ‘Indifferent,’ is associated positively with age (b = 0.06, *p* < 0.01). The number of children is associated positively with both Profile 2/ Negative Attitude (b = 0.29, *p* < 0.01) and Profile 3/ Positive Attitude (b = 0.42, *p* < 0.001). Given that the number of children is examined as an independent variable, one may reason that the parents with more children appear divided, i.e., there is a segment of those parents who perceive the involvement with apps as high and worry about their children’s use of apps and a segment that highly appreciates apps and worries less. Knowledge of new technologies affects positively Profile 3/ Positive Attitude and negatively Profile 4/ Indifferent attitude. That is, parents with the higher knowledge of new technologies greatly appreciate apps most probably belong to Profile 3/ Positive Attitude (b = 0.41, *p* < 0.01) and thus worry less about their use, while parents with the lower knowledge on new technologies most probably belong to the ‘Indifferent’ Profile 4 (b = −0.34, *p* < 0.01). Finally, parents who perceive that the time spent by children on screens is too much most probably belong to Profile 2/ Negative Attitude (b = 0.50, *p* < 0.001), including those who do not appreciate educational apps and perceive high involvement and worry about their children’s use of them. In contrast, parents with mild and indifferent profiles think oppositely (b = −0.23, *p* < 0.01 and b = −0.48, *p* < 0.001, respectively). The level of education does not affect any profile formation.

#### 3.3.2. Association of Profiles with Dependent Variables

The four clusters/profile memberships were considered independent factors associated with distal outcomes, such as the frequency their children use educational apps. Other outcomes were age children started using apps, the perception of a causal relationship between apps use and parent–children conflicts, the overall position on apps use, the possibility that the individual would suggest apps to another parent, and the degree the parent is annoyed by children using apps in public (Table 6).

First, no association with the starting age of children’s app use was found. Profile 2 (Negative Attitude), a parent with negative attitudes and high worries about the apps use, most probably do not use apps often (b = −0.90, *p* < 0.0001), while a parent in Profile 3 (Positive Attitude), with a positive attitude and low worries, most probably do (b = 0.81, *p* < 0.0001). Additionally, parents in Profile 2 (Negative Attitude), contrary to the other profiles, do consider that the apps use causes conflicts between parents and children to a higher degree (b = 0.61, *p* < 0.0001). An overall positive position for apps use is strongly exhibited by Profile 3 (b = 1.13, *p* < 0.0001) and less by Profile 1 (b = 0.43, *p* < 0.01), while the Profile 2 exhibits negative overall position (b = −1.27, *p* < 0.001). Similarly, parents in Profile 3 (Positive Attitude) most probably would recommend apps use to other parents (b = 1.23, *p* < 0.001), and so would parents with Profile 1 (b = 0.62, *p* < 0.001), while parents with Profile 2 (Negative Attitude) would not (b = −1.39, *p* < 0.001). Finally, parents who are annoyed by children’s use of apps in public are those with Profile 2 (Negative Attitude) (b = 0.44, *p* < 0.001).

## 4. Discussion

### 4.1. Psychometric Properties of the PEAU-p

Focusing on the research questions concerning PEAU-p, it was shown that the proposed new instrument, designed to measure parents’ perception of apps used by kindergarten children, possesses satisfactory psychometric properties. The factorial validity demonstrated by CFA meets adequate fit, while the values of Cronbach’s alpha for each factor are by far satisfactory, ensuring high reliability. The six factors structure: Usability, Enjoyment, Involvement, Learning, Worries, and Values, is interpretable, including the dimensions upon which parents can base their mindset and appraise the use of educational apps. The above dimensions are implicitly or explicitly encountered in the relevant literature. Many researchers have highlighted that educational apps should contain a learning goal and be mentally activating, engaging, socially interactive, contain meaningful learning [81,82]. Guided by the evidence-based assumption that parental engagement in technology use is by far influential [53,83], the present study has provided insights for (a) the properties that parents identify in apps for young children, (b) how parents appraise those features, and (c) how those features of apps are in line with parents’ anticipations and needs when they consider them for use by their children. The PEAU-p is an instrument available to researchers for investigating those relationships and comprises a valuable tool for the continuous contribution to theory building in this field. Our work extended previous endeavors by establishing that young children’s parental perceptions regarding digital media use are a multi-dimensional construct [53].

### 4.2. Parents’ Profiles Regarding Perceptions about Apps Use

Having designed a valid and germane apparatus, a methodological choice was made to further explore the latent variables under study by implementing a person-centered approach instead of an alternative variable-centered approach to investigate potentially formative attitudes of parents. In this approach, the basic assumption, from the psychometric theory point of view, is that the observed pattern of responses in a set of questions is due to the existence of latent variables that are categorical in nature. In other words, given a set of criteria or attributes, individuals can be identified as members of distinct groups, possessing comparable levels of those attributes. This categorization process needs robust statistical tools, such as the LCA implemented in this endeavor, to reveal significantly different groups/clusters corresponding to interpretable profiles. This part of the analysis answered the second research question and showed four parents’ profiles, characterized by different attitudes toward the apps’ use. We found that parents with distinct profiles may have opposing perceptions and, thus, they are expected to make different decisions for their children. Literature reports that discovering parents’ beliefs about children’s apps enhances our understanding of parents’ decision-making process in selecting apps for their children [50]. The present classification scheme, resulting from LCA, is the first one reported in the field, which, quantitatively and qualitatively, documented parents’ viewpoints potentially affect children and shape their appropriate screen viewing from early childhood. Determining a typology of parents’ perceptions and farther their beliefs and attitudes on apps, one assesses their set of coherent judgments that could be characterized along the lines of the current debate on young children’s apps use in home settings. The revealed latent profiles can serve as a causal interpretation of the observed parents’ behavior and choices reported in the relevant literature. The results of this research are in line with the findings of other studies, where it has been shown that parents have specific mindsets and needs to satisfy when selecting apps for their children [4,34,44,46,65,84,85]. They desire apps fostering independent entertainment [86], encouraging coeducation between parents and children, and providing personalized and challenging content to their child [44]. When it comes to young children, some issues arise regarding the suitability and the potential profits of using these technologies with educational apps.

The appropriate technology matching with children’s developmental needs can help their cognitive growth and learning outcomes, especially when families control the issue and actively play their supportive role. It presupposes, of course, sensibly designed and attentively applied technology for gaining the benefits of accelerating, amplifying, and expanding the anticipated impact [87]. Parents’ attitudes and actions regarding apps use are active and decisive, forbidding or allowing and encouraging children to use apps in their everyday lives. Parents often pursue the apps that primarily entertain and encourage their children to be autonomous, considering that they would favorably respond to this involvement. It was reported that participants with at least one child aged between 3 to 7 years old required apps developmentally compatible with the child’s age, his/her actual needs with exceptional design and personalized content [34].

### 4.3. Individual Differences on Parents’ Profiles

Parents’ perceptions and attitudes on app use by children are affected by a multiplicity of individual differences. For instance, research reports have indicated that parent education and socioeconomic status (SES) are related to app consumption habits. Children who watch more television and play more video games most likely belong to families with lower SES and parents with low education levels. These differences and the digital gap are being ameliorated, given that these technologies with promising learning benefits are globally increased [39,88,89].

### 4.4. Contribution to the Research Field

The present report adds to our knowledge on those potentially influential factors by revealing some other individual differences. This is related to the third research question concerning the association of the ensued parents’ profiles with other variables, acting as covariates or distal outcomes. The latent classes in question, as dependent variables, are correlated to parents’ age, number of children, and parents’ knowledge level on new technologies. Moreover, the latent profiles as independent variables significantly affect the frequency of apps use, perceptions of conflicts between parent-child, the overall position on apps use, the possibility of recommending apps use to other parents, and the degree of annoyance children’s use of apps. Of course, in the above association, the arrow of causality is defined by the researcher, and, besides the worth of knowing them for theoretical and practical purposes, they are critical because they add to validity considerations, as far as the psychometric measurement and the classification procedure, is concerned, since all of them are meaningful, establishing reasonable and interpretable relationships. To this end, the present work contributes to the relevant empirical research literature with an increasing interest that comprises a growing area of investigations.

### 4.5. Limitations and Future Directions

The present study has some limitations that constrain the implications of the findings. The study uses cross-sectional data, which did not evaluate whether the relations demonstrated here are sustained over time. Future research could investigate the stability of these relations during the elementary school years through longitudinal studies. In addition, the findings cannot be generalized to a specific population since the sampling process did not cover detailed geographic areas and demographic characteristics. Another limitation might concern the data collection procedure, which was conducted via an online questionnaire.

Additionally, it is noteworthy that the vast majority were mothers, which prevented exploring gender differences among parents. Finally, the study was correlational and did not allow firm causal inferences. Based on the current work and other studies in the literature, one might design interventions aimed explicitly at selected aspects of the models presented here, e.g., interventions on parental stereotypes or parents’ beliefs and attitudes about apps.

The proposed tool to measure parental perceptions and the methodology implemented can serve as valuable assets in subsequent inquiries probing the respective perception of parents and the differences between them regarding varying attitudes and behaviors as far the use of educational apps. In the future, it would be interesting to investigate the extent to which the parenting styles could be associated with variations in individual needs. For instance, if authoritative or permissive parenting style impacts parents’ perceptions and attitudes and how they affect young children on the issues addressed in the paper. The ensued parental profiles may be studied with other individual differences, while interventions or longitudinal studies, probing mechanisms of change and overcoming resistance to using educational apps, could be designed and carried out in the light of the present findings.

## 5. Conclusions

The present endeavor adds to the research area on technology and childhood by investigating parental perceptions and attitudes on early childhood educational apps. The contribution of this work is primarily on theory building since it explored, via the designed and validated instrument, PEAU-p, the latent variables determining the mediating role of parents, and provided interpretation of their actions and practices. Providing the optimal learning environment for their children is a challenge for many families. While it is acknowledged that educational apps can help children build academic competence, such as literacy, numeracy, and social skills, and, in the end, to become more prepared for school education, parents do not always conform with this evidence-based postulate. There are counterviews and opposing opinions of parents, preventing the appropriate choices and practices from being fostered. Given the plethora of factors, problems arise from their inability to distinguish the educational value of an app.

Moreover, issues originate from personal worries and doubts. LCA attained a classification of those characteristics, and since it is a model-based method, the ensued latent classes represent specific trends that could be generalized in the population. According to the present findings, only 20.5% of the participants were positively aligned with the apps’ implementation (Cluster/Profile 3). A total of 19.13% of the sample is indifferent and apathetic towards educational apps (Cluster/Profile4). In addition, 31.31% of the sample shows a mild (positive) attitude (Cluster/Profile 1), while 29.07% of the sample is totally against (negative attitude profile) to educational apps (Cluster/Profile 2). Excluding indifferent Profile 4, which would require a differentiated approach to be understood, Profiles 2 and 4 attract the primary attention. In Profile 2 (negative attitude), parents express doubts about possible harm due to hardware technology and high involvement with psychological implications.

On the other hand, Profile 1, with mild attitudes, could be viewed as representing an intermediate or transition state between Profiles 2 and 3. An emergent question and potential hypothesis divide those tendencies, and how to shift between stages could be attained. The strength of the various perceptions demarcates the differences between profiles set by the classification method. At this point, it is pertinent to highlight that those perceptions, most likely, are formed by or due to misinformation, lack of knowledge, and dysfunctional beliefs that operate under bounded rationality [90]. These crucial determinants of the underlying mindsets regarding potential changes and shifts in attitudes and behavioral patterns could be approached and explained by the bounded rationality theory [91].

The theoretical and practical implications derived from the findings of this study contribute to the relevant research field by providing early childhood researchers with both an enriched theoretical premise to interpret empirical data and a valid instrument to capture parents’ perceptions, which probably change over time [92,93]. Additionally, educational organizations might use the current findings to develop programs to inform parents about educational apps by preschool children, emphasizing the specific issues raised by each profile. This is especially important, as screen time and technological devices have increased due to the demands of the current period of the COVID-19 pandemic, establishing new behaviors and habits for young children. Finally, parents, being aware that using developmentally appropriate technology at home promotes young children’s early development and learning, should re-examine their decisions and choices based on firm understanding and knowledge about new technologies embedded formally or informally in contemporary education.

## Figures and Tables

**Figure 1 behavsci-11-00082-f001:**
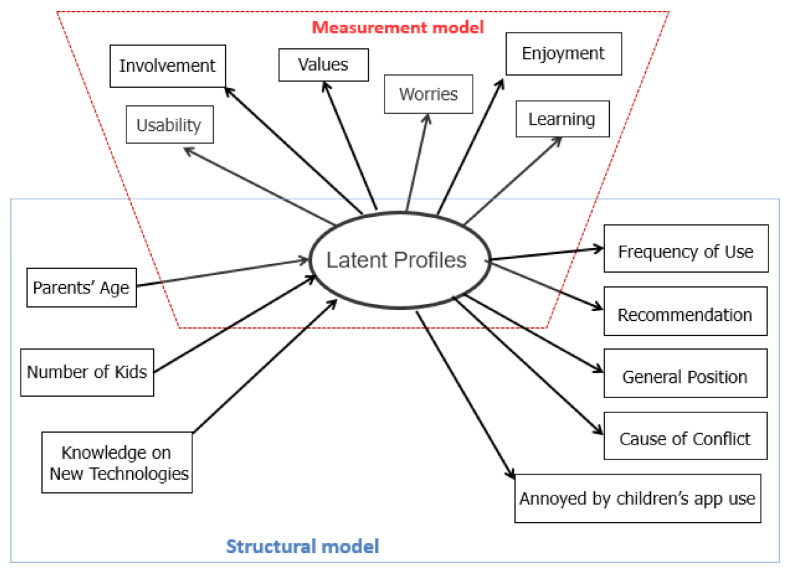
The latent variable model with the measurement and the structural parts.

**Figure 2 behavsci-11-00082-f002:**
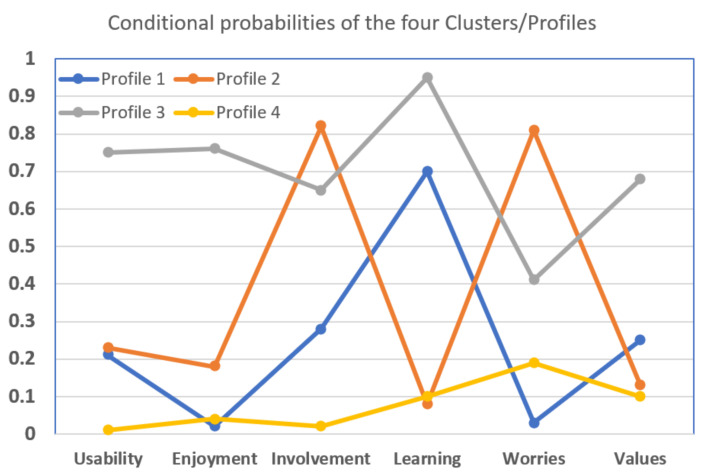
Profiles of the four clusters/Profiles in terms of the conditional probabilities are depicted on the vertical axis, while the horizontal axis shows the six dimensions of the PEAU-p scale.

**Table 1 behavsci-11-00082-t001:** Structure Factors of PEAU-p. Results from PCA with Varimax rotation.

		Factors					
		LE	W	EN	VA	US	IN
Learning Outcomes	F4-Q17. Facilitate new knowledge acquisition.	0.776					
	F4-Q18. Enhance a child’s language development.	0.761					
	F4-Q21. Promote creative thinking.	0.751					
	F4-Q16. Contribute to the cognitive development	0.747					
	F4-Q19. Facilitate foreign languages learning	0.740					
	F4-Q20. Offer feedback in case of error.	0.723					
	F4-Q15. Promote logical thinking.	0.681					
Worries	F5-Q28. Undermine children development		0.890				
	F5-Q29. They create health problems.		0.844				
	F5-Q27. Problems are due to radiation.		0.837				
	F5-Q30. Reduces quality interaction with parents.		0.798				
	F5-Q31. Cause introversion in children.		0.773				
Enjoyment	F2-Q06. They offer pleasant sounds.			0.881			
	F2-Q05. They enclose pleasant images.			0.851			
	F2-Q08. They contain fun characters for kids.			0.850			
	F2-Q07. They entertain the children.			0.727			
Values	F7-Q37. They are complemented by traditional teaching.				0.890		
	F6-Q38. They offer multimedia teaching material.				0.868		
	F6-Q39. They strengthen the motivation for learning.				0.783		
	F6-Q36. They create an effective learning environment.				0.544		
Usability	F6-Q03. They provide instructions suitable for children of this age.					0.775	
	F1-Q01. They are easy to use by children.					0.757	
	F1-Q04. Children easily understand the content.					0.721	
	F1-Q02. Children can use them without the guidance of an adult.					0.655	
Involvement	F3-Q12. They capture the child’s attention.						0.833
	F3-Q11. They capture the child’s interest.						0.800
	F3-Q14. They create an addiction to the child.						0.580

**Table 2 behavsci-11-00082-t002:** Correlation matrix of the six factors, Cronbach’s α, and a number of items.

	US	EN	IN	LE	W	VA	Alpha	Items
Usability	1						0.782	4
Enjoyment	0.478 **	1					0.898	4
Involvement	0.345 **	0.308 **	1				0.768	3
Learning	0.343 **	0.496 **	0.103 *	1			0.896	7
Worries	0.028	−0.112 *	0.379 **	−0.301 **	1		0.889	5
Values	0.279 **	0.406 **	0.111 *	0.575 **	−0.212 **	1	0.882	4
Mean	4.57	5.24	5.88	4.32	4.96	4.31		
SD	1.08	1.18	1.04	1.11	1.33	1.33		

**Table 3 behavsci-11-00082-t003:** Latent Class Analysis. LCA.

	LL	BIC(LL)	Npar	L^2^	df	*p*-Value	Class.Err.	Entropy R^2^
1-Cluster	−2699.08	5471.04	12	922.07	422	0.00	0.00	1
2-Cluster	−2566.25	5284.32	25	656.40	409	0.00	0.0956	0.68
3-Cluster	−2514.06	5258.89	38	552.02	396	0.02	0.1461	0.68
4-Cluster	−2471.57	5252.87	51	467.05	383	0.25	0.1320	0.80
5-Cluster	−2450.55	5289.78	64	425.02	370	0.34	0.1606	0.74
6-Cluster	−2439.45	5346.52	77	402.80	357	0.38	0.1643	0.73

**Table 4 behavsci-11-00082-t004:** A verbal description of the four Clusters/Profiles.

	Cluster 1/Profile 1	Cluster 2/Profile 2	Cluster 3/Profile 3	Cluster 4/Profile 4
	31.31%	29.07%	20.50%	19.13%
Usability	Low	Low	High	Low
Enjoyment	Low	Low	High	Low
Involvement	Medium	High	High	Low
Learning	Medium	Low	High	Low
Worries	Low	High	Medium	Low
Values	Low	Low	High	Low
	Mild Attitude	Negative Attitude	Positive Attitude	Indifferent Attitude

**Table 5 behavsci-11-00082-t005:** Effects of covariates with the ensuing parents’ profiles.

Covariates	Profile 1/Mild Attitude	Profile 2/Negative Attitude	Profile 3/Positive Attitude	Profile 4/Indifferent Attitude
Parent Age	0.02	−0.075 ***	−0.01	0.06 **
Number of Children	−0.31	0.29 **	0.42 ***	−0.39
Age of the 1st Child	0.00	0.02	−0.01	−0.01
Age of the 2nd Child	0.06	−0.05	−0.11	0.09
Knowledge of New Technologies	−0.12	0.05	0.41 **	−0.34 **
Perceived crucial time on Apps	−0.23 **	0.50 ***	0.21	−0.48 ***
Level of Education	−0.05	0.02	0.20	−0.17

* *p* < 0.05. ** *p* < 0.01. *** *p* < 0.001.

**Table 6 behavsci-11-00082-t006:** Effects of parents’ Profile-memberships on distal outcomes.

Dependents	Profile 1/Mild Attitude	Profile 2/Negative Attitude	Profile 3/Positive Attitude	Profile 4/Indifferent Attitude
Frequency of use	0.33 *	−0.90 ***	0.81 ***	−0.24
Child’s age started	0.03	−0.05	0.00	0.01
Cause of conflicts	−0.59 ***	0.61 ***	0.21	−0.23
Overall positive Position	0.43 **	−1.27 ***	1.13 ***	−0.29
Possibility of recommendation	0.62 ***	−1.39 ***	1.23 ***	−0.46 **
Annoyed by children’s use of apps	−0.19	0.44 **	−0.15	−0.13

* *p* < 0.05. ** *p* < 0.01. *** *p* < 0.001.

## Data Availability

Not applicable.

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
