# Peer review of "Parents’ Perceptions of Educational Apps Use for Kindergarten Children: Development and Validation of a New Instrument (PEAU-p) and Exploration of Parents’ Profiles"

_behavsci, 2021, doi:10.3390/bs11060082_

Round 1
Reviewer 1 Report
Congratulations, an excellent research. A fantastic article
Author Response
Respond to reviewers.
Reviewer 1
Comments and Suggestions for Authors
Congratulations, an excellent research. A fantastic article
ANS: Thanks
Reviewer 2
The state of art mentions aspects such as the role of parents in the children use of educational applications, as well as their perception.
OK Thanks
However, a research question is posed related to psychometric characteristics that are not mentioned previously.
ANS: OK. , it was added in the abstract and in the main text –in the passage before Methods
I believe that an analysis on these issues is lacking, or else the purpose of the study should not focus on them. In fact, the conclusions place more emphasis on the educational aspect of the applications.
ANS: OK the analysis the psychometric properties, that is related to validity and reliability issues covers one aspect of the paper and it is prerequisite for conducting the empirical research, In conclusions indeed we place emphasis educational aspect because the implications concern education.
The study is interesting but the objectives achieved could be better differentiated by relating them to the results more precisely.
ANS: OK corrected
In Figure 1 (p. 9) it would be convenient not to abbreviate Number of Kids.
ANS: OK corrected
In the P. 10 the colors of Profiles 3 and 4 are not indicated.
ANS: OK corrected. (Thanks)
Reviewer 3
This manuscript is written well, however, this need to be revised a few points.
Abstract:
Results of this study need to show here more exactly.
ANS: OK an attempt was made to include all aspect and findings.
Introduction:
Authors show that "The present study aims to investigate this contemporary issue and contributes by presenting an instrument to measure parents’ perceptions on apps and further exploring the role of selected individual differences. However, describe the hypothesis of this study more specifically is needed.
ANS: OK an attempt was made to revised this passage.
Results: With regard to Fig. 2, Please explain the contents of Fig. 2 more concretely. I think the explanation of the text and figures is a little insufficient.
ANS: OK the caption of Fig 2 and the corresponding text provides complete description.
Discussion: The author needs to be more specific about the limitations of this study. Are there any other research restrictions?
ANS: OK more limitations are provided.
Reviewer 4
This paper presents a very interesting and actual issue. It reveals originality when carrying out a tool to measure parents perception about the use of technologies by their children. The text is very well structured, with a clear language and without spelling errors. The statistical analysis is appropriate and the results are clearly presented. Discussion is also well organized and conclusion is based on the findings. In terms of the subject the paper presents a balanced argument, showing that are people favour to the use of technologies by young children as well as people against it. An innovation is that the parameters for these opinions are clear here and a tools can be used for measure these arguments.
ANS: Thanks
Reviewer 2 Report
The state of art mentions aspects such as the role of parents in the children use of educational applications, as well as their perception. However, a research question is posed related to psychometric characteristics that are not mentioned previously. I believe that an analysis on these issues is lacking, or else the purpose of the study should not focus on them. In fact, the conclusions place more emphasis on the educational aspect of the applications.
The study is interesting but the objectives achieved could be better differentiated by relating them to the results more precisely.
In Figure 1 (p. 9) it would be convenient not to abbreviate Number of Kids.
In the P. 10 the colors of Profiles 3 and 4 are not indicated.
Author Response

(The authors gave the same response as above.)

Reviewer 3 Report
This manuscript is written well, however, this need to be revised a few points.
Abstract:
Results of this study need to show here more exactly.
Introduction:
Authors show that " The present study aims to investigate this contemporary issue and contributes by presenting an instrument to measure parents’ perceptions on apps and further exploring the role of selected individual differences. However, describe the hypothesis of this study more specifically is needed.
Results: With regard to Fig. 2, Please explain the contents of Fig. 2 more concretely. I think the explanation of the text and figures is a little insufficient.
Discussion: The author needs to be more specific about the limitations of this study. Are there any other research restrictions?
Author Response

(The authors gave the same response as above.)

Reviewer 4 Report
This paper presents a very interesting and actual issue. It reveals originality when carrying out a tool to measure parents perception about the use of technologies by their children. The text is very well structured, with a clear language and without spelling errors. The statistical analysis is appropriate and the results are clearly presented. Discussion is also well organized and conclusion is based on the findings. In terms of the subject the paper presents a balanced argument, showing that are people favour to the use of technologies by young children as well as people against it. An innovation is that the parameters for these opinions are clear here and a tools can be used for measure these arguments.
Author Response
Reviewer 1
Comments and Suggestions for Authors
Congratulations, an excellent research. A fantastic article
ANS: Thanks
Reviewer 2
The state of art mentions aspects such as the role of parents in the children use of educational applications, as well as their perception.
OK Thanks
However, a research question is posed related to psychometric characteristics that are not mentioned previously.
ANS: OK. , it was added in the abstract and in the main text –in the passage before Methods
I believe that an analysis on these issues is lacking, or else the purpose of the study should not focus on them. In fact, the conclusions place more emphasis on the educational aspect of the applications.
ANS: OK the analysis the psychometric properties, that is related to validity and reliability issues covers one aspect of the paper and it is prerequisite for conducting the empirical research, In conclusions indeed we place emphasis educational aspect because the implications concern education.
The study is interesting but the objectives achieved could be better differentiated by relating them to the results more precisely.
ANS: OK corrected
In Figure 1 (p. 9) it would be convenient not to abbreviate Number of Kids.
ANS: OK corrected
In the P. 10 the colors of Profiles 3 and 4 are not indicated.
ANS: OK corrected. (Thanks)
Reviewer 3
This manuscript is written well, however, this need to be revised a few points.
Abstract:
Results of this study need to show here more exactly.
ANS: OK an attempt was made to include all aspect and findings.
Introduction:
Authors show that "The present study aims to investigate this contemporary issue and contributes by presenting an instrument to measure parents’ perceptions on apps and further exploring the role of selected individual differences. However, describe the hypothesis of this study more specifically is needed.
ANS: OK an attempt was made to revised this passage.
Results: With regard to Fig. 2, Please explain the contents of Fig. 2 more concretely. I think the explanation of the text and figures is a little insufficient.
ANS: OK the caption of Fig 2 and the corresponding text provides complete description.
Discussion: The author needs to be more specific about the limitations of this study. Are there any other research restrictions?
ANS: OK more limitations are provided.
Reviewer 4
This paper presents a very interesting and actual issue. It reveals originality when carrying out a tool to measure parents perception about the use of technologies by their children. The text is very well structured, with a clear language and without spelling errors. The statistical analysis is appropriate and the results are clearly presented. Discussion is also well organized and conclusion is based on the findings. In terms of the subject the paper presents a balanced argument, showing that are people favour to the use of technologies by young children as well as people against it. An innovation is that the parameters for these opinions are clear here and a tools can be used for measure these arguments.
ANS: Thanks